# Automated Categorization of Multiclass Welding Defects Using the X-ray Image Augmentation and Convolutional Neural Network

**DOI:** 10.3390/s23146422

**Published:** 2023-07-14

**Authors:** Dalila Say, Salah Zidi, Saeed Mian Qaisar, Moez Krichen

**Affiliations:** 1Hatem Bettaher Laboratory, IResCoMath, University of Gabes, Gabes 6029, Tunisia; 2CESI LINEACT, 69100 Lyon, France; 3Electrical and Computer Engineering Department, Effat University, Jeddah 22332, Saudi Arabia; 4Faculty of Computer Science and Information Technology, Al-Baha University, Al-Baha 65528, Saudi Arabia; 5ReDCAD Laboratory, National School of Engineers of Sfax, University of Sfax, Sfax 3038, Tunisia

**Keywords:** CNN, deep learning, multi-class classification, data augmentation, welding defects, segmentation

## Abstract

The detection of weld defects by using X-rays is an important task in the industry. It requires trained specialists with the expertise to conduct a timely inspection, which is costly and cumbersome. Moreover, the process can be erroneous due to fatigue and lack of concentration. In this context, this study proposes an automated approach to identify multi-class welding defects by processing the X-ray images. It is realized by an intelligent hybridization of the data augmentation techniques and convolutional neural network (CNN). The proposed data augmentation mainly performs random rotation, shearing, zooming, brightness adjustment, and horizontal flips on the intended images. This augmentation is beneficial for the realization of a generalized trained CNN model, which can process the multi-class dataset for the identification of welding defects. The effectiveness of the proposed method is confirmed by testing its performance in processing an industrial dataset. The intended dataset contains 4479 X-ray images and belongs to six groups: cavity, cracks, inclusion slag, lack of fusion, shape defects, and normal defects. The devised technique achieved an average accuracy of 92%. This indicates that the approach is promising and can be used in contemporary solutions for the automated detection and categorization of welding defects.

## 1. Introduction

Welding plays an important role in various industries, including automobile, aerospace, petrochemicals, mechanics, electrical, etc. Due to the welding process’s complexity, and the welding parameters’ instability, welding flaws are predictable, for example, cracks, lack of fusion, and cavity [1]. Thus, verifying the welds’ quality is essential to ensure the welding morphology’s safety [2]. Non-destructive testing is an active procedure and a principal technique to detect the inside quality of diverse weld joints. For the remarkable precision welding segment, in demand for assessing the destructiveness of underage imperfections, the goal is not solely to identify the presence of flaws but also to pinpoint their exact location [3]. There are several NDT methods, such as the penetrant test, acoustic emission (AE), visual test (VT), infrared thermography test (IRT), ultrasonic test (UT), and radiographic test (RT), used to uncover welding flaws inside pipelines. The identification of thin flaws in welded joints with traditional methods is still a major problem in the industry, along with the waste of time, materials, and employment. Further, the results can be poor and require the repetition of the operation several times. In this context, several researchers have used deep learning to eliminate ancient problems and improve experimental results. The UT method is used with the implementation of a deep learning algorithm to unconsciously detect and find the geometric components of welds [4], then, it is applied to detect the crack welding defect in a gas pipeline [5]. The AE [6] control technique is employed to identify and categorize the diverse weld flaws inside carbon steel. In addition, the integrity of the welds can be inspected by NDT methods with the radiography test based on the X-ray radiation passing through a specimen to detect any defects in the joint. X-ray radiation is employed for slender materials, though gamma rays are applied for materials. The findings of the testing can be evaluated using a range of methods, including computed radiography, film, and CT scans [7,8]. Currently, these films are scanned to be treated on a numerical computer. Subsequently, scanned images are categorized by little contrast and badly off quality; inspecting weld defects can be difficult. Unfortunately, these qualities affect explanation and classification. This categorization is a procedure of classification, where they are identified and sorted accordingly. Admittedly, the categorization of images is becoming a popular area of computer vision and is expected to be performed in the upcoming years [9]. Figure 1 presents some X-ray weld defect images.

Usually, this type of recommendation is founded on three stages: preliminary preparation, subdivision and characteristic extraction, and categorization. Normally, the characteristic extraction phase is frequently performed by experts and hence is a long and imprecise process. Certainly, this accuracy issue is mainly due to the variety of weld defect types. Further, the generated features are insufficient for the recognition quality to detect errors effectively [9,10]. Following, many studies have examined the general structure of machine learning applications and the algorithms used for feature extraction and ML. However, some artificial neural networks have simple applications, specialized classification, and low accuracy. In recent times, there has been a notable rise in the utilization of deep learning methodologies, driven by their ability to leverage extensive datasets and perform automated feature extraction. These capabilities have proven to be highly valuable in the efficient detection of defects within welds. Defect prediction plays a pivotal role in evaluating the condition of a weld, and deep learning techniques have emerged as prominent and effective tools for this purpose in welding processes. By harnessing the power of deep learning, practitioners can significantly enhance their ability to predict defects, leading to more accurate and reliable assessments of weld quality [11]. This integration of deep learning techniques into welding processes holds great potential for improving overall welding performance and ensuring the integrity of critical components and structures. Thus, computer vision and ML have been created to support experts in evaluating the outcomes, with the use of an innovative method that combines laser profilometry and machine learning algorithms to automatically detect defects in welds by utilizing high-throughput 3D data acquisition and lightweight machine learning techniques [12].

We aim to develop a model that is appropriate for our situation and can facilitate the automated processing of the intended dataset with an appropriate level of accuracy. The importance of the welding defect inspection is to improve the welding quality and decrease the fault identification degree produced by human influences, such as incorrect detection and incompetent manual examination. Researchers have suggested a diversity of weld flaw identification approaches founded on computer vision, which can make effective defect detection intelligent and standardized [7,8,13]. For this reason, we are brought to work with this tendency industrial vision. Further, we use a particular deep-learning architecture for welding verification with specific data that remains challenging. Furthermore, to control the limitations of the preceding work and participate with our dataset consisting of six types of defects—lack of fusion, cavity, cracks, inclusion, shape defects, and normal defects—we use the X-ray images to evaluate the NDT method for the weld imperfections. The approach of data augmentation results in an increase in the dataset size, while simultaneously decreasing the occurrence of overfitting. However, we apply the CNN model to achieve the classification of the six defects, then the results are presented in a confusion matrix and curve form. Figure 2 presents some X-ray weld defect images that we use to create our new dataset—based on the public “GDxray” dataset [14].

In many of the previous studies, as discussed in the related works section, the CNNs were utilized with precise goals in mind. However, several researchers have devised new and innovative methods to address the limitations of CNNs from dissimilar viewpoints.

This work aims to automate and improve the precision of the multiclass categorization of welding defects in X-ray images. To achieve this, we present several contributions to this study:The automated and precise multi-class categorization of welding defects in X-ray images.Creation of a new dataset by segmenting and augmenting the original “GDXray” dataset, the Grima Database of X-ray images that contains multiple welding defects [14]. The newly generated dataset serves as the foundation for our automated categorization approach. In the future, it can be processed and explored by other potential researchers.Developing a CNN model that directly processes the augmented dataset for the automated multiclass categorization of welding defects in X-ray images.

Section 2 details some research on weld detection and classification with traditional methods, arriving at new models founded on deep learning. Then, in Section 3, welding identification issues, mostly in the dataset, are presented and the moving-on performance for the CNN is inspected and educated in detail. The outcomes are introduced in Section 4, and Section 5 accomplishes our paper.

## 2. Related Works

Image classification is applied to identify weld flaws in X-ray images. The major purpose is to classify radiographic images into different categories that would help localize and evaluate defects. The quality classification is 71% and shows great application value. The methods for detecting weld defects are the subject of extensive research. Shevchik et al. [13] suggested a technique founded on a deep artificial neural network (ANN) that could detect process instability defects in real-time. The quality categorization falls within the range of 71% to 99% and displays a strong commitment to implementing values. Ajimi et al. [9] proposed a new classification data improvement founded on a deep learning network for the random transformation of images on data and used image preprocessing and machine learning tools for traditional crack detection; this method proved that the prototypical delivers exceptional performance within a limited time frame. Additionally, the SVM and the ANN, applying four texture features assigned to defect detection and classification, were judged and then required supplementary performance accuracy [15]. Also, Ajimi et al. [16] developed a method for assessing the performance of deep learning networks across a range of parameter and hyperparameter configurations. They also incorporated an improved learning process into the dataset, which led to a 3% increase in the accuracy of the model. Hou et al. [17] advanced a deep learning model that could directly analyze X-ray images using a neural network. They evaluated the model’s ability to classify features and compared its performance to that of traditional techniques, using dissimilar datasets; the model had 97.2% accuracy, which is much more advanced than feature extraction methods. The model was evaluated using the signal-to-noise ratio, origin mean square error, entropy, and reliability of the model. Nacereddine et al. [18] proposed computer-aided diagnostic software for welding defects and proposed unsupervised classification based on a limited combination model of multivariate generalized Gaussian distribution. Su et al. [19] recognized an automatic identification system of weld-joint defects by extracting texture characteristics from weld defects. According to a new area of ML, deep learning has considerable probability in the area of identifying imperfections, constantly decreasing the measurement in the feature procedure to keep away from the effect of characteristic extraction on the recognition outcomes, thus successfully refining the precision of the flaw recognition. Zhang et al. [20] suggested a transfer learning classification technique for training two deep neural networks on an improved image dataset. The images are classified into different categories using a multi-model addition structure. In [21], the same researcher devised an eleven-layer CNN model for the classification and identification of weld defects. He proposed a non-pixel technique for a quantifiable evaluation and visualization of the model performance. Jiang et al. [22] offered an enhanced pooling approach that accounts for both the pooling region and characteristic map delivery, which leads to a notable increase in identification precision. To further highlight the significance of computer vision, Cardellicchio et al. [12] presented a pioneering method that combines laser profilometry and machine learning algorithms to automatically detect defects in welds. By leveraging high-throughput 3D data acquisition and employing lightweight machine learning techniques, the proposed approach achieves remarkable accuracies of 99.79% for defect identification and 99.71% for defect categorization. This method exhibits significant potential for implementation in real production lines, enabling cost reduction, real-time monitoring, and the efficient evaluation of defects. The prediction of weld defects is important for ensuring high-quality and safe welded structures. Baek et al. [11] proposed a method based on deep learning to predict weld penetration depth in arc welding, demonstrating its potential for real-time monitoring and the prediction of weld quality using surface images of the weld pool. The developed prediction model achieved high accuracy, with a mean absolute error of 0.0596 mm and an $R^{2}$ value of 0.9974, utilizing semantic segmentation and regression techniques.

In summary, the integration of deep learning, artificial intelligence, and specialized methodologies, such as X-ray radiography, molten pool extraction, and 3D data analysis, presents noteworthy advantages in enhancing the precision and automation of weld classification, detection, and quality control processes. These approaches yield notable benefits, including precise and automated defect classification, the real-time monitoring of weld quality, improved defect recognition through advanced strategies, and even the real-time prediction of weld penetration depth. Nevertheless, it is essential to acknowledge the substantial resource requirements associated with training data, computational capacity, initial investments, and technical expertise. Consequently, the meticulous consideration and thoughtful implementation of automatic weld quality control methods must incorporate these factors to ensure optimal outcomes and cost-effectiveness in industrial applications.

## 3. Methodology

In our study, the challenge of the curse of dimensionality emerged due to the scarcity of training data. To surmount this obstacle, we employed two primary approaches: data augmentation and modifications to the model’s architecture. Data augmentation involved applying diverse transformations to the existing dataset, effectively expanding our training set artificially. This technique fostered dataset diversity, mitigated overfitting risks, and improved the model’s ability to generalize. Simultaneously, we streamlined the model’s complexity by reducing the number of layers and parameters. This architectural modification aimed to counteract overfitting tendencies, enabling the model to better accommodate the limitations of the available data and enhance its generalization capabilities. The combination of these strategies enabled us to effectively address the curse of dimensionality, optimize model training, and achieve superior results, despite the constraints imposed by our limited dataset. In our research, we implemented transfer learning and fine-tuning techniques specifically tailored to a convolutional neural network (CNN) model for the classification of various types of weld defects in images. By adapting the model to our specific task, our objective was to enhance its accuracy and efficacy in identifying and categorizing different types of weld flaws. Although we did not conduct explicit evaluations of advanced layers, such as residuals or global average pooling, we explored alternative models beyond the initial implementation of AlexNet. Through our experiments, we assessed the performance of CNN, VGG16, random forest, and Inception V3 models, all of which exhibited significantly improved results compared to AlexNet. As a result, we redirected our focus toward these models for further analysis and refinement.

### 3.1. Dataset

In this research, we use a public experimental X-ray inspection database (GDXray dataset) created for research and educational purposes. The dataset of primary X-ray inspection includes a section of X-ray weld images that were gathered by BAM, the Federal Institute for Materials Research and Testing in Berlin, Germany. The subset of welds consists of 10 X-ray images that are 4K in length and have varying widths, known as the W0001 series. Additionally, this subset includes explanations of bounding cases and binary images that represent the actual data for weld flaws, referred to as the W0002 series. The Welds W0003 sequence now includes more than 67 X-ray images to better support network training [14].

The welding process has numerous kinds of imperfections (ISO6520-12007) [23]. The key categories of welding imperfections are cavity, crack, inclusion slag, lack of fusion, shape defects, and dimensions. Certain approaches deal with the identification of these defects without categorizing them. However, the proposed approach not only deals with the identification of these defects but also provides a deeper insight by categorizing the welding defects.

Traditionally, X-ray inspection focuses on sub-superficial defects. However, in this study, superficial defects such as cracks are also included. We believe that such an approach provides valuable insights into the potential challenges and risks associated with the welding process. The intended categories of defects are cracks, cavities, solid inclusion, shape defects, lack of fusion, and normal defect (image without defect).

The crack defects are the rectilinear defectiveness of welds. These are produced primarily when the internal welding stresses surpass the mechanical strength of the filler metal, the base metal, or both. It is a crucial goal in the fabrication process. Among all welding imperfections, cracks are considered the most severe and are generally deemed unacceptable as per manufacturer specifications. It should be noted that cracks may not always manifest immediately after welding but can gradually develop due to cyclic fatigue loads during service. Forces like tension, bending, torsion, and shear, as well as thermal expansion and contraction, have the potential to generate cracks long after the completion of welding. Hence, it is essential to effectively manage welding stresses and employ appropriate measures to minimize the occurrence of cracks in welds. In order to effectively mitigate the occurrence of crack defects in welds, it is of paramount importance to employ X-ray techniques for the early detection of such flaws in joints prior to any extension processes. The cavity’s imperfections are produced by stuck gases or contractions. Holes of gases considered by a circular shape are the most shared. Then, the solid inclusions are non-metallic particles in the welded metal or at the interface. In addition, the lack of fusion is the absence of a union between the metal and base metal or between layers of the weld metal. Furthermore, the term “normal defect” in X-ray images is the digital representation that presents an object without any modification or undesirable visual flaws. These images present the true essence of the subject without misrepresentations, abnormalities, noise, and/or other imperfections. The standardized ISO5817:2014 [24] offers an example of receipt heights for weld imperfections frequently used in manufacturing as the base for additional case-specific recognition standards. Figure 2 shows the six classes of welding defects.

To evaluate our research, firstly a new dataset is created by applying manually a cropping technique of the original X-ray images. Additionally, a data augmentation method uses elevating the image quantity and splitting it to a train set and test set to train a CNN model to classify the six defects weld.

### 3.2. Data Augmentation

Data augmentation is an important step before training and testing the intended learning model [9,24,25]. The employed data augmentation steps are described below:Random rotation: This technique in the valves rotates an image by a certain number of degrees around its original center point. The degree of rotation can be either to the right or left and can range from 1 to 359 degrees. In many cases, the rotation process, which is symbolized as R is combined with zero padding to fill out any missing pixels as it is shown in Equation (Equation 1). We apply a random rotation of 40 degrees:
(1)$$R=\left[\begin{array}{cc} cos(\alpha ) & -sin(\alpha )\\ sin(\alpha ) & cos(\alpha ) \end{array}\right]$$Shearing (shear range): The process of shearing involves applying a transformation, denoted as H, to an image. This transformation moves each point in the image in a particular direction, and the distance of the movement depends on the point’s distance from a line that runs parallel to the selected direction and passes through the origin as it is shown in Equation (Equation 2):
(2)$$H=\left[\begin{array}{cc} 1 & h_{x}\\ h_{y} & 1 \end{array}\right]$$The symbols hx and hy represent the shear coefficient along the x and y axes, respectively. In our case, the shearing is equal to 20%.Zooming: This technique refers to the process of changing the image’s size to enhance its visibility or to focus on a specific part of the image. It involves enlarging or reducing the image’s dimensions. The process S is given by Equation (Equation 3) and can be achieved independently in diverse instructions. We apply a zoom range of 20%:
(3)$$S=\left[\begin{array}{cc} S_{x} & 0\\ 0 & S_{y} \end{array}\right]$$Brightness: This method is a quality of an image that indicates how light or dark it is. This can refer to specific areas within the image or the overall illumination of the scene. The luminosity of the image can be transformed by appending a 0.2 to all pixel ethics. Yet, we apply the brightness range between 0.5 and 1.5.Flips: Flipping is a method of making a reflection from the original image. In a two-dimensional image, the positions of the pixels are flipped or mirrored along one of the axes, either horizontally or vertically. In our case, we use only horizontal flipping.Even though some small changes are made to the images, their important meanings remain the same, and the images are still labeled based on their original training label. In other words, the modifications to the images do not affect their core semantic content or the original classification they were given during training. Figure 3 presents the different data augmentation techniques.

### 3.3. Convolutional Neural Network (CNN) Architecture

In this study, CNN is used for automated categorization of the intended welding defects. Its architecture contains four kinds of layers, which are input, convolution, pooling, and fully connected, corresponding to the hidden layers of a CNN model [26,27].

One key advantage of the CNN is that it avoids the feature-extraction step in image pre-processing and simplifies the classification process [26,28,29]. The CNN reduces the intended images to a more trainable form, without losing the essential information which is required for the classification [30]. These layers are described below. Figure 4 shows a global architecture of a CNN.

The layers are pre-prepared by fixing their number and types [31,32]. The input layer presents the data settings (width, high, size, and channel) to the model. The convolutional layer relates a usual of several separators applied to the images and executes a linear function to the input. The filters extract features from the image and create an equivalent map for each item. The convoluted data are then handled by an exchange capability named rectified linear unit (ReLU) [33,34]. The ReLU presents non-linearity in the network; subsequently, the greatest of the data in the CNN method is non-linear. The attribute maps, generated by the convolving and ReLU, are the input to the pooling layers [32,35,36].

The pooling layer intends to rebuild samples of the feature map output. The concept is to put on a filter and a stride of a similar distance on the convolutional layer output. The precise character from the authentic input is recognized, and the comparative position near other features is more important than its required position [23,35]. This principle of this layer is depicted with the help of Figure 5.

The fully connected (FC) layer indicates that each junction in the preceding layer is allied to each neuron in the presentation layer. To have the option to interface each hub on the principal fully connected layer to the previous coat, the results of multi-layered exhibits should be placed in a singular cluster. It is completed by putting on vectorization to the matrix to achieve a linear transformation to a one-row vector. Each neuron in the FC coat takes completely the output from the past layers as input [16,35].

The last CNN layer is the output. It is the last hidden layer that provides the predictions of the network. The activation function applied in this stage is the softmax function. This function bandages the output from the last hidden layer to probability values in all classes. Consequently, the combination of the outputs is continuously equivalent to 1. The output of the classification layer agrees with the class with the maximum probability [9,21].

### 3.4. The Evaluation Metrics

One of the most common issues that training algorithms often encounter is the need to prevent overfitting. In this study, regularization is used to overcome the overfitting issue [16]. To measure the learning performance of the CNN model, the confusion matrix is utilized as a conventional evaluation technique. In the field of image categorization, a confusion matrix is applied to evaluate its predicted results to real values and measure its degree of performance. Generally, by using the confusion matrix, we cannot just identify accurate and inaccurate predictions, yet more importantly, we can gain insight into the specific types of errors that are made. To calculate a confusion matrix, one needs a set of test data and another of validation, which contains the values of the obtained results. The attribution of six types of welding flaws can be immediately recognized by employing this traditional method. The evaluated measure is illustrated in Figure 6.

The results are sorted into four groups:True positive (TP): The “true positive” in a confusion matrix is the number of occurrences where the model accurately predicts the positive class or event among all true positives in the dataset.True negative (TN): in a confusion matrix, “true negative” refers to the number of instances where the model correctly predicts the negative class out of all the true negative instances in the data.False positive (FP): The “false positive” represents the error made by the model when it wrongly predicts the presence of a specific condition or event, even though it is not present in reality. This type of error is also known as a Type I error, and it can lead to incorrect decisions if not properly managed.False negative (FN): A “false negative” in a confusion matrix is when the model incorrectly predicts the absence of an event among all actual positive instances in the data. It is a measure of the model’s tendency to miss positive cases or make Type II errors.

The harmonic mean of precision and recall is provided by the F1-score. The number of samples of the true response that belong to an intended class is known as the support. The support in the realm of data analysis and machine learning represents the quantitative measure of the frequency or occurrence of an item, event, or class within a given dataset. It encapsulates the prevalence of an item set or the significance of a class by quantifying their respective frequencies in the dataset. Through the assessment of support, insights are derived regarding the distribution of data and the evaluation of patterns or categories within the dataset.

## 4. Results and Discussion

### 4.1. The Dataset Preparation

The community database (GDXray) supplied by BAM Federal Institute for Materials Research and Testing in Berlin, Germany [14], was used as the data source for upcoming experimental studies. This database’s “welding” defects contain 67 defect images of diverse sizes and types.

Using previous knowledge, defects in cropping were identified by manually selecting examples from images showing imperfections. These flaws were then classified and labeled with class labels before being included in the ‘weld defects’ dataset as depicted in Figure 7.

The overview of the suggested process is exposed in Figure 7. The proposed system aims to address the limitations of existing systems by using a balanced dataset with X-ray defect images from the GDXray dataset. The GDXray dataset contains various types of weld defects, such as cavities (C), cracks (CR), inclusion (I), lack of fusion (LOF), shape defects (SDs), and normal defects (NDs). The first step in the proposed system is the manual segmentation of X-ray images to collect a new database. This segmentation process helps in identifying and isolating the regions of interest (ROIs) containing the weld defects, which are then used for further analysis. After segmentation, the technique of data augmentation is applied to increase the size of the database. Data augmentation techniques, such as rotation, scaling, flipping, and adding noise, can be used to artificially create more training samples, thereby enhancing the volume and diversity of the dataset. The dataset is then split into training (80%) and test sets (20%), which are used for training and evaluating the CNN architectures for defect classification. The CNN models are trained using the training dataset, and the accuracy and the loss are computed after each epoch to monitor the model’s performance during training. Validation loss and accuracy are also calculated to evaluate the model’s performance on the unseen data from the test set. The overall system performance is measured using a confusion matrix, which provides insights into the model’s classification performance for each class. Accuracy, precision, recall, and F1 score are also calculated to assess the system’s performance in terms of correctly identifying the weld defects. Figure 8 likely depicts the flowchart or schematic of the proposed system, illustrating the steps involved in the defect classification process.

### 4.2. Proposed CNN Model

The proposed CNN model used in the classification of the X-ray image defects; it is shown in Table 1.

The proposed model has five 2D convolutional layers, five ReLU activation layers, five pooling layers, and one dropout layer. The dropout rate that is added to the model is a method of regulation in the neural network that supports decreased interdependent learning between the neurons. Dropout mentions overlooking units through the training stage of a confident set of neurons, which is selected at random. The first convolution layer creates a 32-feature map, the second convolution layer makes 64 characteristic maps, the third and the fourth have 128 feature maps, and finally, 128 feature maps are generated at the last convolution layer. Python programming language is used for model creation with Keras–TensorFlow as the backend. Figure 9 represents the proposed model for weld defect detection. By applying to fine-tune the model parameters, for the proposed model, the first convolutional layer has a kernel size of 5 × 5, then the other four convolutional layers have convolutional filter size 3 × 3, pooling filter size 2 × 2 and a dropout rate of 0.5.

Adam is the optimizer applied in two diverse sets of research. They are set with a training rate equivalent to 0.001, a batch size equivalent to 32, and several epochs equal to 100 epochs. Followed by that, checkpoints are introduced to serialize the model which has the greatest validation accuracy. Later, the finest model weights can be protected from being troubled while training. Lastly, the model is trained on 3452 images and tested through 867 images that are arbitrarily chosen from the dataset.

### 4.3. Discussion

In this topic of research, the multiclass classification method is designed to detect different weld defects from the public dataset “GDXray” based on a CNN model. To devise the model, all images are resized to 128 × 128. Additionally, the total dataset is divided into training (80%) and testing (20%). Multiclass classification methods use a CNN model, where the hyperparameters are applied to improve the accuracy of the weld defect classification; for this algorithm, we use an epoch size equal to 100 and batch sizes equal to 16 to advance the accuracy of several weld flaw detections. After that, mixing the greatest optimizer (Adam) and a loss function is a categorical cross-entropy with a multiclass classification; using the CNN model attained a better accuracy of 92%.

The utilization of a convolutional neural network (CNN) model for weld defect classification is well supported by scientific and formal justifications. CNNs have a strong track record in image-based classification tasks, attributed to their capacity to extract relevant features and capture spatial relationships. This makes them a suitable choice for accurately identifying and classifying weld defects. Additionally, fine-tuning the hyperparameters of the CNN model ensures optimal performance specific to weld defect classification, enabling the capture of distinct defect characteristics. The selection of an epoch size of 100 and a batch size of 16 strikes an effective balance between computational efficiency and model optimization. Leveraging the Adam optimizer facilitates efficient parameter updates, particularly advantageous for CNN models dealing with large parameter spaces. Moreover, the application of the categorical cross-entropy loss function effectively minimizes classification errors in multiclass classification scenarios. Collectively, these considerations contribute to the CNN model’s ability to deliver accurate and precise weld defect classification outcomes. The value is utilized for an extra validation procedure and applied as a predictor. The training and validation curve of the algorithm is presented in Figure 10 and Figure 11.

Due to the variety of welding defects in the considered dataset, the accuracy score alone may not provide a reliable evaluation of the performance. Therefore, other known evaluation measures such as precision, recall, and F1-score, are also considered in this study. The performance evaluations are summarized in Table 2.

Many parameters can influence the outcomes of a CNN model, including the number of convolutional layers, the kernel size, and the number of filters. Other important parameters include the learning rate, batch size, activation function, and pooling strategy. Each of these parameters can impact the performance of the CNN in various ways; optimizing them is critical for achieving the best accuracy.

We aim to create a convolutional neural network (CNN) capable of accurately classifying the different types of welding defects. To achieve this goal, we need to make a careful choice of the parameters of the CNN model. It includes the choice of the number of convolutional layers, the number of filters in each layer, and the size of the kernel used in the filters. The size of the kernel refers to the dimensions of the convolutional filter used to scan the input data; it can be 3 × 3, 5 × 5, or 7 × 7. These parameters play a critical role in determining the performance of the CNN and can significantly impact the model’s accuracy. To make well-informed decisions about these parameters, we conduct a thorough review of the existing literature. We incorporate empirical evidence to choose values that strike a balance between model complexity and performance. Ultimately, our goal is to develop a CNN architecture that would effectively and efficiently classify images for our particular task.

Although the training and validation results show stagnation after approximately 30 epochs, there are scientific justifications for continuing the network training. Persisting with the training, the process holds the potential of achieving incremental enhancements in the model’s performance, despite the improvements being modest. Extending the training duration enables researchers to investigate diverse learning rates, regularization techniques, and model architectures to optimize the model’s performance. Furthermore, prolonging the training phase allows for the assessment of potential overfitting and the implementation of suitable strategies to mitigate its impact. This comprehensive approach ensures a thorough exploration of the model’s capabilities, leading to maximized accuracy and improved generalization capabilities.

Figure 12 presents various outcomes that suggest that modifying the number of convolutional layers and filters in a CNN can affect the model’s accuracy, complexity, and computational demands. Increasing these parameters can improve the ability to extract features and classify data while decreasing them can lead to a simpler model with fewer computational requirements but potentially lower accuracy and feature extraction capabilities. Selecting the best parameters relies on the data complexity and the preferred balance between model performance and computational efficiency. Based on the findings in Figure 12, the model attains 92% accuracy by using a 5-layer convolutional approach with 32 filters and a 5 × 5 kernel size. These parameters are determined through experimentation to achieve a desirable balance between accuracy and computational requirements.

Table 2 presents the summary of evaluation metrics, used to evaluate the performance of the proposed model. The choice of these evaluation measures is made based on their extensive usage in literature for evaluating the performance of welding defect classifiers.

The precision measures the proportion of true positives (correctly predicted positive samples) out of all positive predictions (true positives and false positives), while recall measures the proportion of true positives out of all actual positives. Using both precision and recall can provide a more accurate understanding of the model’s performance, especially in situations where there is a class imbalance in the data.

For a more comprehensive evaluation of the model’s performance, the F1 score can be calculated based on precision and recall. The F1 score is a weighted average of precision and recall and is particularly useful in situations where both precision and recall are important for the problem being solved (the parameters are summarized in Figure 6).

In our research, we conduct an investigation using multiple models, namely InceptionV3, random forest (RF), LeNet, and CNN, to address our research objectives. Following rigorous simulations and experiments, we carefully evaluate and compare the performance of these models. Among the evaluated models, the CNN model emerges as the most promising and delivers excellent results. This finding suggests that the CNN architecture possesses advantageous characteristics that enable it to effectively handle the complexities and nuances of the dataset. To provide a comprehensive and quantitative representation of our findings, we intend to present the detailed results in a tabular format. Table 3 displays the various accuracy results of the models.

In our research, we integrate the Optuna optimizer to perform hyperparameter tuning. Optuna is an advanced optimization framework that utilizes intelligent search algorithms and Bayesian optimization techniques to effectively explore the hyperparameter space and discover the optimal configurations. Through leveraging Optuna’s capabilities, our objective is to identify the most suitable values for hyperparameters, like batch size, number of epochs, and others, which significantly impact the learning process of our model. This optimization process results in an impressive learning rate of 91% after conducting 20 trials, indicating the exceptional performance of our model. To provide a comprehensive summary of our findings, Table 4 presents the various hyperparameter values.

Table 5 displays various accuracy results obtained through research on welding defect classification in X-ray images. The studies conducted demonstrate that the CNN method proposed in our research paper exhibits better accuracy than the method used in their previous work. In recent years, several research studies have been conducted to classify welding defects in X-ray images using different methods. The accuracy achieved by each technique or model depends on several factors, such as the complexity of the problem, the size and quality of the dataset, the features extracted, and the choice of the algorithm used. For instance, Jian et al. [24] used the SVM method to classify six welding defects and achieved an accuracy of 90.4%. Also, Celia Cristina et al. [32] used Alexnet to identify also six defects and achieved an F1-score of 70.7%. Meanwhile, a deep neural network (DNN) pre-trained by the SAE method, used by Yang and Jiang [15], seems to have achieved the highest accuracy among the mentioned studies, with an accuracy of 91.36%. Hou et al. [17] used the histogram of oriented gradients (HOG) technique to classify five defects, achieving an accuracy of 81.60%.

HOG is a popular feature extraction technique that captures the gradient information in an image, which can be useful in differentiating between different defects. Furthermore, Kumaresan et al. [25] used a convolutional neural network (CNN) to identify ten defects and achieved an accuracy of 89%. Finally, Dong et al. [37] used random forest to classify four defects with an accuracy of 80%. Random forest is an ensemble learning algorithm that combines multiple decision trees to improve accuracy and reduce overfitting.

Our main contribution to this research is the creation of a new dataset through manual segmentation of the original X-ray images. This process allows us to extract the relevant portions of the images that contain welding defects, resulting in a more focused and representative dataset for training our CNN model. This approach not only improves the accuracy of our model but also ensures that the training data are more robust and diverse, thus making them more suitable for real-world applications. For more comprehension, the creation of a new dataset for training a CNN model can be beneficial in many ways. For example, it can lead to better accuracy by providing more relevant and representative data, which can reduce the number of false positives and false negatives in the model’s predictions. Additionally, a new dataset can increase diversity by including a wider range of variations, such as different types and sizes of welding defects, which can improve the robustness of the model. By overcoming the limitations of the original dataset, a new dataset can also provide customization by selecting specific types of welding defects or adjusting the size and resolution of the images to optimize the model’s performance. Finally, a new dataset can offer reusability for future projects by serving as a foundation for other similar applications or projects, thus saving time and resources.

Based on the methodology we employ, including the creation of a new dataset through manual segmentation and the use of data augmentation and fine-tuning strategies, our CNN model achieves a significantly better accuracy rate of 92% in detecting welding defects in X-ray images. This result indicates that our approach is successful in improving the performance of the model and overcoming the limitations of previous studies that used different methods, such as SVM or Alexnet. Therefore, we can confidently state that our model achieves better results, which can have important implications for the detection and prevention of welding defects in various industries. The CNN approach is more effective and has important implications for future research.

The the incorporation of the event-driven and adaptive rate approach can be beneficial in terms of compression, computational cost, and latency [38]. Moreover, the optimization-based feature selection can further enhance the precision and computational effectiveness [39]. The feasibility of integrating these methods, in the proposed solution, can be investigated in the future.

## 5. Conclusions

In our research, we perform a multi-class classification method to detect weld flaws in X-ray images by employing a CNN model. The model’s creation relies on GDX-ray image databases that are publicly available. This database contains only a minimum number of samples. Therefore, we extract the sub-images and create our dataset, which is used for the training and test process. The technique of data augmentation and fine-tuning strategies helps to achieve the appropriate performance of the model. The multi-class classification is combined with the optimizer algorithm. The model achieves an accuracy score of 92% for the case of the CNN classifier. This performance is comparable or superior compared to its counterparts. The applicability of the proposed method is only tested for the intended dataset. In the future, the performance of the proposed approach will be studied for extended datasets in order to achieve the automated categorization of weld defects. Also, its application to other industrial problems like packaging inspection, defect detection in fabric and textiles, and printed circuit board inspection will be investigated. Another future work is to investigate the possibilities of enhancing the precision of our approach by incorporating feature extraction, feature selection and ensemble learning techniques.

## Figures and Tables

**Figure 1 sensors-23-06422-f001:**
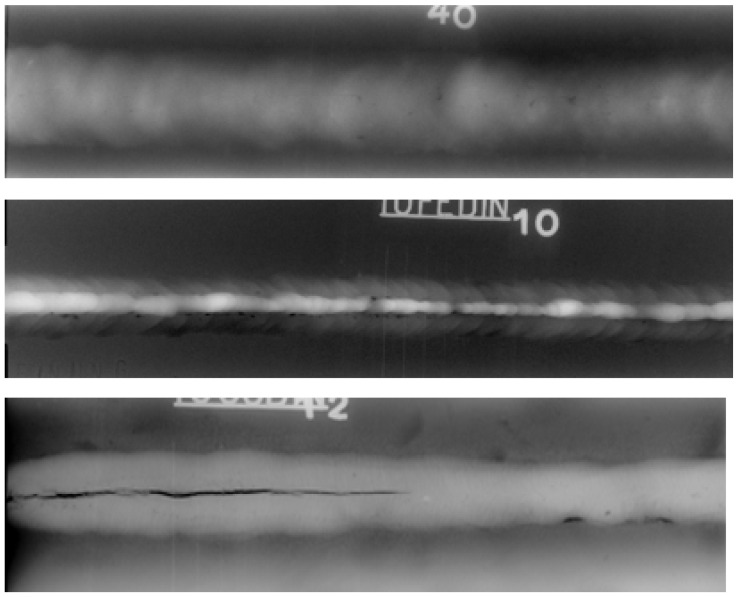
X-ray weld defect images.

**Figure 2 sensors-23-06422-f002:**
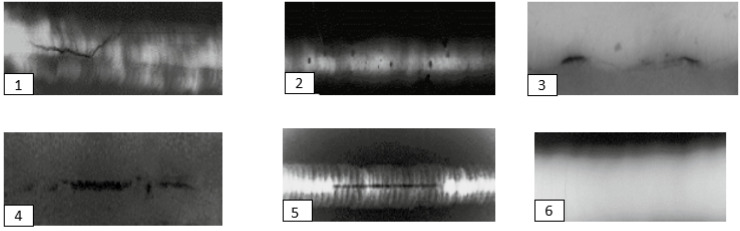
Weld defects: 1 = cracks; 2 = cavity; 3 = shape defects; 4 = inclusion; 5 = lack of fusion; 6 = normal defect.

**Figure 3 sensors-23-06422-f003:**
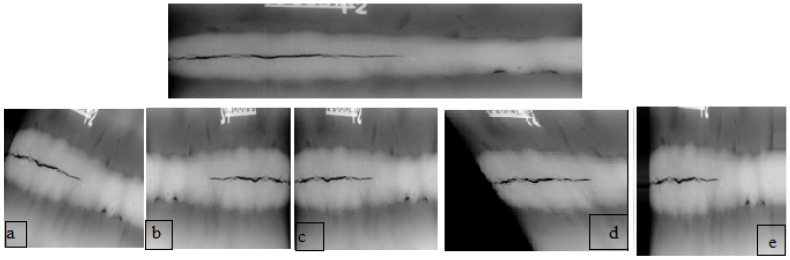
X-ray images data augmentation techniques: (**a**) rotation, (**b**) horizontal flipping, (**c**) brightness, (**d**) shear, and (**e**) zoom.

**Figure 4 sensors-23-06422-f004:**
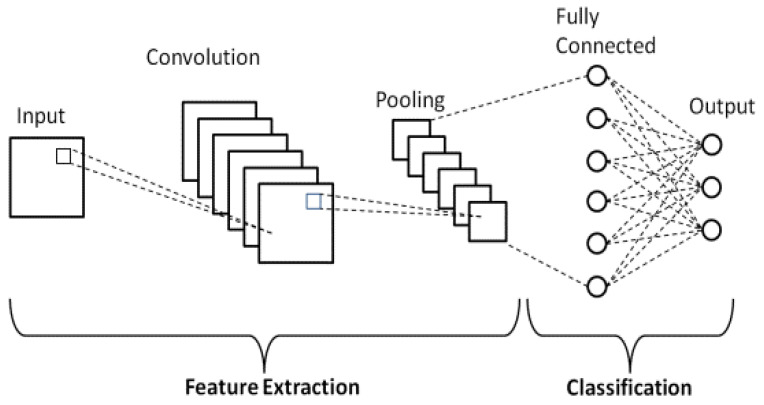
CNN architecture.

**Figure 5 sensors-23-06422-f005:**
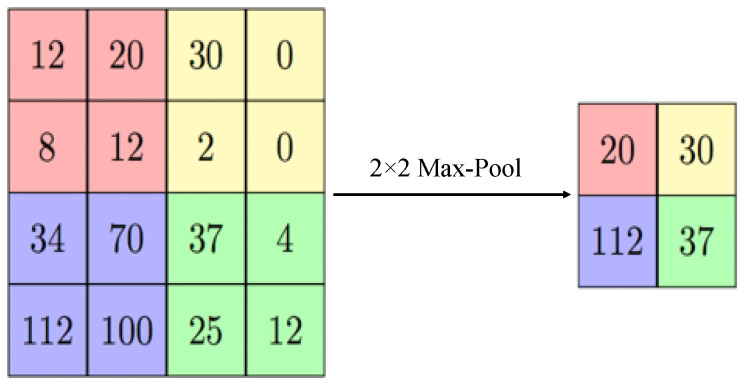
Example of the max-pooling with a 2 × 2 image.

**Figure 6 sensors-23-06422-f006:**
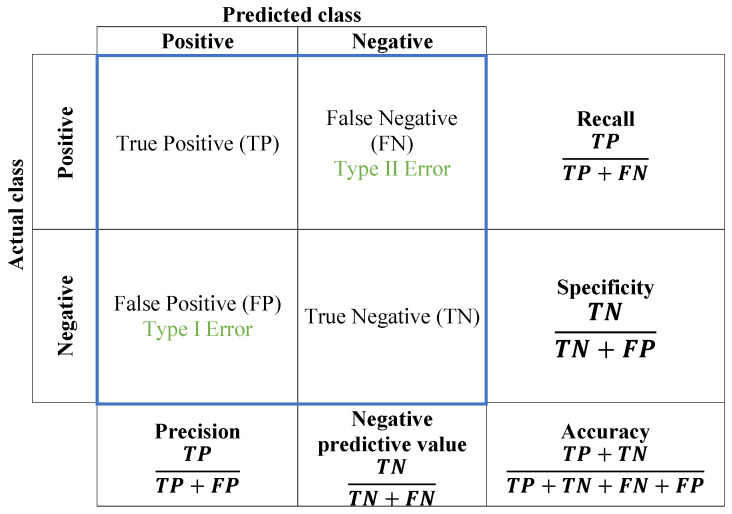
Performance evaluation of a confusion matrix.

**Figure 7 sensors-23-06422-f007:**
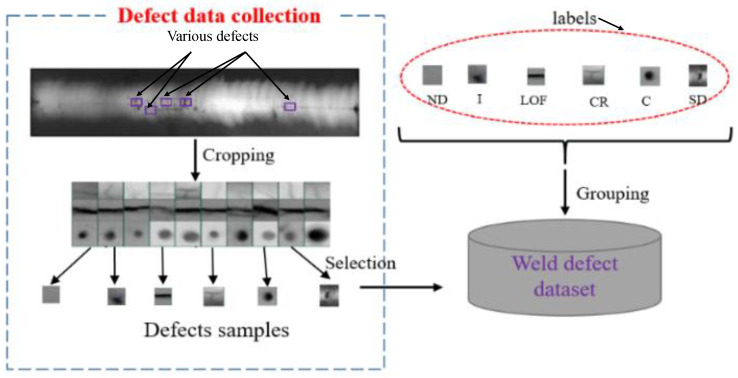
Founding of the weld defect dataset.

**Figure 8 sensors-23-06422-f008:**
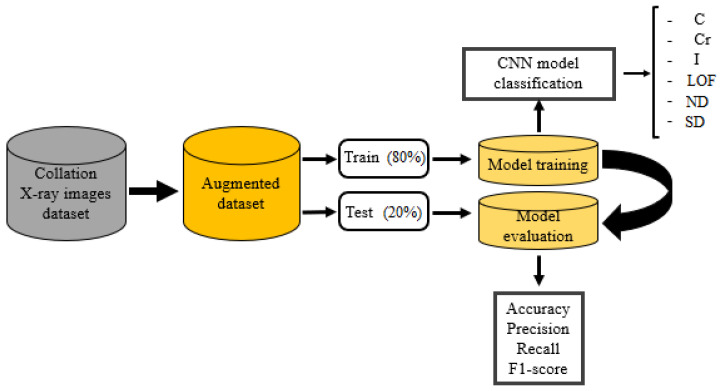
The proposed method of the CNN classification.

**Figure 9 sensors-23-06422-f009:**
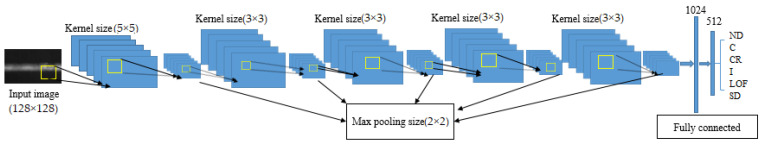
Designing a convolutional neural network (CNN): proposed architecture.

**Figure 10 sensors-23-06422-f010:**
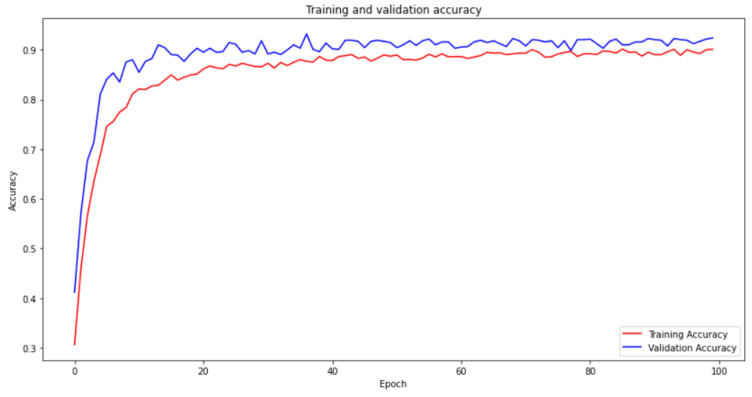
Accuracy curves for training and validation over 100 epochs.

**Figure 11 sensors-23-06422-f011:**
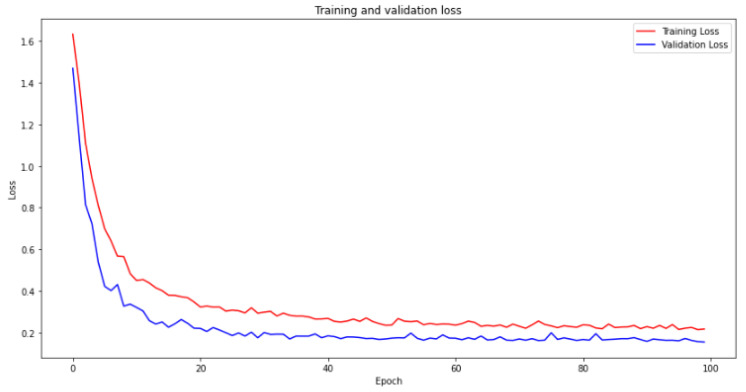
Loss curve comparison between training and validation phases over 100 epochs.

**Figure 12 sensors-23-06422-f012:**
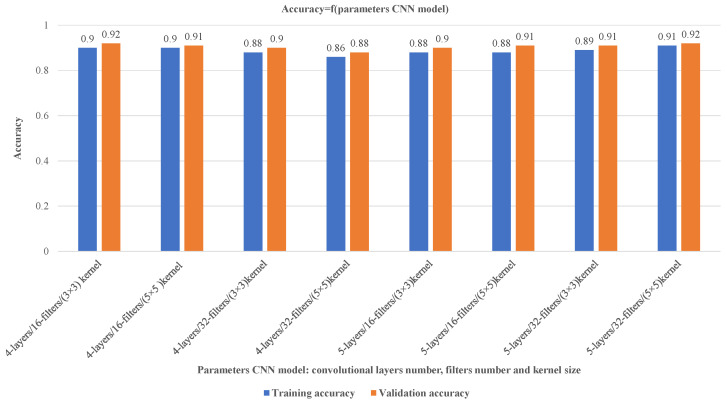
Optimizing CNN model parameters for improved training and validation accuracy.

**Table 1 sensors-23-06422-t001:** Summary of the proposed CNN model.

Layer (Type)	Output Shape	Param #
conv2d (Conv2D)	(None, 124, 124, 32)	2432
max_pooling (MaxPooling2D)	(None, 62, 62, 32)	0
conv2d_1 (Conv2D)	(None, 60, 60, 64)	18,496
max_pooling2d_1 (MaxPooling2D)	(None, 30, 30, 64)	0
conv2d_2 (Conv2D)	(None, 28, 28, 128)	73,856
max_pooling2d_2 (MaxPooling2D)	(None, 14, 14, 128)	0
conv2d_3 (Conv2D)	(None, 12, 12, 128)	147,584
max_pooling2d_3 (MaxPooling2D)	(None, 6, 6, 128)	0
conv2d_4 (Conv2D)	(None, 4, 4, 256)	295,168
max_pooling2d_4 (MaxPooling2D)	(None, 2, 2, 256)	0
dropout (Dropout)	(None, 2, 2, 256)	0
Flatten (Flatten)	(None, 1024)	0
Dense (Dense)	(None, 512)	524,800
Dense_1 (Dense)	(None, 6)	3078

**Table 2 sensors-23-06422-t002:** Evaluation metrics.

	A Model Trained and Compiled Using Adam			
	Precision	Recall	F1-score	Support
**C**	0.97	0.79	0.87	98
**CR**	0.99	0.93	0.96	163
**I**	0.85	0.87	0.86	162
**LOF**	0.89	0.93	0.91	163
**ND**	0.99	0.99	0.99	120
**SD**	0.87	0.97	0.91	161
**Accuracy**			0.92	867
**Macro avg**	0.93	0.91	0.92	867
**Weighted avg**	0.92	0.92	0.92	867

**Table 3 sensors-23-06422-t003:** Accuracy model’s results.

Model	Accuracy
InceptionV3	88.58%
RF	86%
LeNet	85%
CNN	92%

**Table 4 sensors-23-06422-t004:** Optuna optimizer.

	Filters	Kernel_Size	Units	Dropout_Rate	Batch_Size	Accuracy
Optuna optimizer	32	4	448	0.28	32	91%

**Table 5 sensors-23-06422-t005:** Comparison of different weld defect detections.

Method	Number of Defect Classes	Accuracy
**DNN pre-trained by SAE** [15]	5	91.36%
**HOG** [17]	5	81.60%
**CNN** [25]	10	89%
**SVM** [24]	6	90.4%
**AlexNet** [32]	6	F1-score (70.7%)
**Random Forest (RF)** [37]	4	80%
**Proposed method**	6	92%

## Data Availability

The data presented in this study are openly available. For complete details, interested authors may refer to the article [14] with doi: https://doi.org/10.1007/s10921-015-0315-7.

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
