# Peer review of "Automated Categorization of Multiclass Welding Defects Using the X-ray Image Augmentation and Convolutional Neural Network"

_sensors, 2023, doi:10.3390/s23146422_

Round 1
Reviewer 1 Report
This paper describes a method that uses CNNs to perform defect detection on weld images acquired using X-rays. Here are my suggestions to improve the overall quality of the paper:
1. The authors focus only on techniques that use CNNs for image classification on X-ray images. However, recent works have also used other computer vision techniques to extract several traits from welds, using for example 3D reconstruction and ML methods for classifying superficial defects. Hence, I think the first section can be improved by including more relevant recent works, such as:
a. https://doi.org/10.1007/s10845-023-02124-1
b. https://doi.org/10.1007/s10845-022-02013-z
I also suggest further improving the state-of-the-art section by providing more details on the advantages and disadvantages of the enlisted approaches. This gives the reader a proper perspective on the motivation underlying the authors’ proposal.
2. What do you mean by “normal defect” (Figure 2, caption)? Is this a normal sample?
3. As for the defects categories, why did you choose also superficial defects (e.g., cracks)? Usually, X-Ray is focused on sub-superficial defects.
4. The description of the effect of each data augmentation process is too long, and in my opinion provides low value to the reader. Hence, it can be safely reduced. The same goes for the broad description of CNNs in section 3.3.
5. You have about 3.5k images for training a network that has more than 1M parameters. Did you consider the curse of dimensionality? Did you try transfer learning and fine-tuning on standard CNNs? Also, have you tested the model with other advanced layers and blocks (e.g., residuals, batch normalization, global average pooling, etc.)? If not, why?
6. Training and validation results appear to be flat after about 30 epochs. Why did you keep training the network?
7. Did you consider autoML or hyperparameters tuning?
Overall, the achieved results are good. However, I think the authors should provide a more formal explanation of their choices in determining the overall model. Hence, I think the paper can be considered for publication only after a major revision.
1. The first plural person is not suited for scientific papers. Please use the third person.
2. Also, the possessive form is not suited. Please remove possessives.
3. There are some typos, e.g., row 127: “The same researcher in [19]. Drew”.
Author Response
Thank you for your valuable comment. Please check the pdf document for responses.

Reviewer 2 Report
The authors are to be congratulated in attempting to apply state-of-the-art automation techniques like data augmentation and deep learning algorithm in order to identify the different weld defects with an accuracy of ~92%. However, there are some comments (please see the attachment) which need to be addressed before the publication of this article.

There seem to be many English language and formatting errors throughout the manuscript. These are mentioned in the attachment as well.
Author Response

(The authors gave the same response as above.)

Round 2
Reviewer 1 Report
The authors successfully addressed all highlighted issues. Hence, the paper can be considered for publication.